artificial intelligence

cross subject, electroencephalogram emotion recognition, personality first, deep neural network

**Authors for correspondence:**
Zhidan Zhao
e-mail: zzhidanzhao@gmail.com
Dazhi Jiang
e-mail: dzjiang@stu.edu.cn

# Personality first in emotion: a deep neural network based on electroencephalogram channel attention for cross-subject emotion recognition

Zhihang Tian[1,2,†], Dongmin Huang[1,2], Sijin Zhou[1,2], Zhidan Zhao[1,2] and Dazhi Jiang[1,2,†]

[1]Department of Computer Science, School of Engineering, and [2]Key Laboratory of Intelligent Manufacturing Technology (Ministry of Education), Shantou University, Shantou 515063, People's Republic of China

ZZ, 0000-0001-9459-7984

In recent years, more and more researchers have focused on emotion recognition methods based on electroencephalogram (EEG) signals. However, most studies only consider the spatio-temporal characteristics of EEG and the modelling based on this feature, without considering personality factors, let alone studying the potential correlation between different subjects. Considering the particularity of emotions, different individuals may have different subjective responses to the same physical stimulus. Therefore, emotion recognition methods based on EEG signals should tend to be personalized. This paper models the personalized EEG emotion recognition from the macro and micro levels. At the macro level, we use personality characteristics to classify the individuals' personalities from the perspective of 'birds of a feather flock together'. At the micro level, we employ deep learning models to extract the spatio-temporal feature information of EEG. To evaluate the effectiveness of our method, we conduct an EEG emotion recognition experiment on the ASCERTAIN dataset. Our experimental results demonstrate that the recognition accuracy of our proposed method is 72.4% and 75.9% on valence and arousal, respectively, which is 10.2% and 9.1% higher than that of no consideration of personalization.

## 1. Introduction

Emotion recognition plays an important role in interpersonal communication and human–computer interaction, and the

[†]Contributions: Zhihang Tian and Dazhi Jiang contributed equally to this work.

research of emotion recognition has been developed for decades. Generally, human emotions can be predicted by three methods: non-verbal behaviour methods (such as facial expression recognition, action recognition, etc.) [1], speech behaviour methods (such as text emotion recognition, conversation emotion analysis, etc.) [2], and methods based on physiological signals (such as electroencephalogram (EEG)-based emotion recognition, electrocardiogram-based emotion recognition, etc.) [3]. Because of the complexity of human emotion expression, many emotion recognition methods use physiological signals such as EEG, electrooculogram (EOG), electromyogram (EMG), Galvanic skin response (GSR), respiration and blood pressure etc. Koelstra et al. [4] analysed the mapping relationship between blood volume pressure, respiratory rate, skin temperature, EOG and emotion caused by 40 music videos. Subramanian et al. [5] studied the binary emotion recognition based on the ASCERTAIN dataset, which include physiological characteristics such as GSR, EEG, electrocardiogram (ECG) and facial landmark trajectory (EMO) etc. EEG stands out from many signals because of its high time resolution, large real-time difference and close connection with different emotional states of the human brain. It has been proved that EEG signals can effectively identify different emotions [6–10]. For computational model problems, researchers have proposed many methods and models to recognize emotion through EEG signals [11–15]. Among the numerous methods of EEG emotion recognition, it is worth noting that in recent years, the method based on deep learning has achieved a dominant position in improving the performance of EEG emotion recognition. For example, Zheng & Lu [11] introduced deep belief networks (DBNs) to construct an emotion recognition model based on EEG. Pandey & Seeja [16] proposed a multilayer perceptron neural network for independent emotion recognition. Song et al. [17] constructed graph relationships based on multi-channel EEG data and convoluted the graph to extract features for classification. Li et al. [18] established a new hierarchical spatio-temporal neural network based on brain regions for EEG emotion recognition. In addition, more and more scholars pay attention to the research of EEG channel distribution and channel selection algorithm which are closely related to emotion. Ansari-Asl et al. [19] proposed a channel selection algorithm based on the synchronization likelihood method. Five channels were selected from 64 emotional EEG channels, and the classification effect was not significantly reduced when identifying positive, medium and negative emotional states. Zhang et al. [20] report a channel selection algorithm based on ReliefF and applied it to the classification of four emotional states: pleasure, fear, sadness and ease. To consider the influence of personality on emotion, the personality theory widely accepted by scholars is the Big Five personality model proposed by Lew Goldberg in 1990 [21]. They believe that human personality can be described in five dimensions—openness, conscientiousness, extraversion, agreeableness and neuroticism (OCEAN). Vinciarelli & Mohammadi [22,23] have twice carried on the comprehensive elaboration to the personality computation. Stemmler & Wacker [24] focused on personality, emotion and individual differences in physiological responses. Winter & Kuiper [25] conducted extensive research on the relationship between personality and emotion in social psychology.

Although some progress has been made, previous works have only considered personality as an undifferentiated, insignificant and independent model feature and do not consider the internal connection between personality and physiological signals. In social psychology, the relationship between personality and emotion has been widely studied [26–28]. In social groups, different people have different cognition and personality, and they may have different reactions under the same emotional stimulus [29]. More importantly, the existing research does not further explore the influence of personality on emotional response. As mentioned above, we have reason to believe that personality is a good indicator to measure the difference of human emotional response under the same emotional stimulus [30]. Hence, we present a deep neural network based on EEG channel attention (DNNECA) model, which attaches great importance to the role of personality in emotion analysis from both the macro and micro points of view comprehensively. Basically, from a macro point of view, personality is taken as the corresponding emotional characteristic of each subject under the same emotional stimulus, and the participants are divided into different groups by calculating the differences between personality. This division process is intuitive and adaptable. Therefore, according to this idea, we apply clustering algorithm to classify the subjects according to their personality. Then explore the influence of personalization on emotion recognition. Specifically, we employ a deep learning model to extract the temporal and spatial feature information of EEG and reveal that the contribution of different EEG channels to emotion recognition is different from the micro perspective. Finally, in order to better analyse personalized emotions, we introduce a channel weight layer, which can establish the internal connection mapping between EEG channels and EEG emotions to highlight the role of EEG channels in EEG emotion recognition.

The remainder of this paper is organized as follows: In §2, we specify the data preprocessing, data structure construction and the deep learning model based on EEG channel attention combined with

**Table 1.** The details of the ASCERTAIN dataset.

| attribute | description |
|---|---|
| subject | 58 college students (21 female, average age = 30) |
| stimulant | 36 movie clips |
| number of trials | 36 |
| length of each trial | 51–127 s |
| recorded EEG signals | 8 electrode and 32Hz sampling rate |
| rating scales | a seven-point scale was used with a −3 to 3 scale for valence, a 0 to 6 scale for arousal, engagement (did not pay attention—totally attentive), liking (I hated it–I loved it) and familiarity (never seen it before–remember it very well) |
| EEG data | trials*channels*data |
| label | trials*labels (arousal, valence, liking, engagement and familiarity) |

personality first (PF). In §3, we conduct extensive experiments to evaluate the proposed method for EEG emotion recognition. Finally, in §4, we discuss and conclude the paper.

# 2. Material and methods

## 2.1. Dataset

To the best of our knowledge, ASCERTAIN [5] is the only released and published dataset that links personality and emotional state through physiological reactions, as shown in table 1. Fifty-eight college students (21 female, average age = 30) who are fluent in English and frequent visitors to Hollywood movies were invited to watch 36 movie clips of 51–127 s from [31]. The movie clips are shown to be uniformly distributed (nine clips per quadrant) over the arousal-valence (AV) plane. During video viewing, several commercial sensors are used to record physiological signals. After watching each segment, participants were asked to label the AV ratings reflecting their affective impression with a seven-point scale, i.e. −3 (very negative) to 3 (very positive) scale for V, and 0 (very boring) to 6 (very exciting) scale for A. Personality measures for big-five dimensions were also compiled using a big-five marker scale questionnaire [32].

## 2.2. Data preprocessing

In order to understand the impurities and noise in the original EEG data in the ASCERTAIN dataset, the dataset needs to be preprocessed. First, for the EEG record segments with data loss and record failure, we remove them directly. Then, the common average reference is applied to the original EEG signals. Next, each EEG electrode signal is filtered with 1 Hz high-pass filter and 16 Hz low-pass filter. Finally, the EOG and EMG artefacts are removed from the EEG data by independent component analysis (ICA).

## 2.3. Data structure construction

Adapt to the proposed EEG emotion recognition model properly, this section reconstructs the representation structure of the EEG signal based on the previous section of EEG signal preprocessing. Generally, for a group of experimental EEG signals, the data are recorded from different EEG channels, which is the data matrix of $M \times L$, where $M$ is the number of EEG channels and $L$ is the length of EEG record. Therefore, a set of two-dimensional step windows with width of $\omega$, height of $M'$ and step length of $s$ is used to cut the data matrix to obtain $U$ group data units. Figure 1 presents

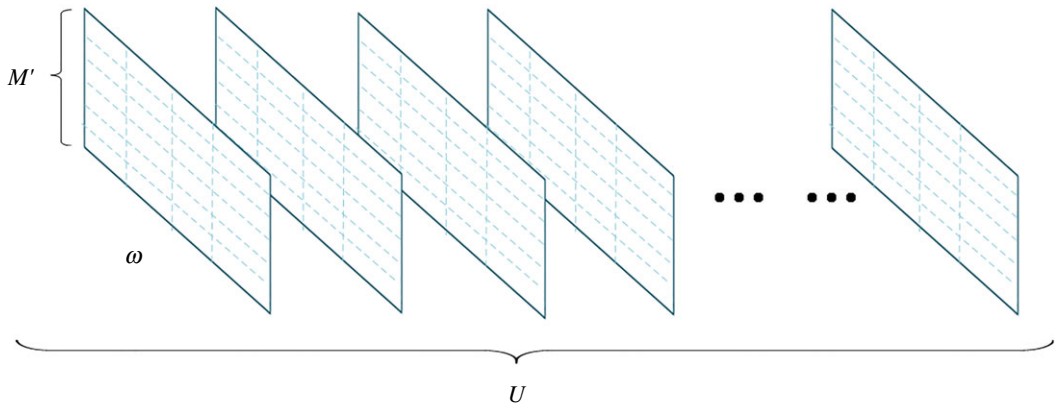

**Figure 1.** Representation structure of EEG data reconstructed into data unit.

the structure after data cutting, where $U$ can be calculated as follows:

$$\begin{cases} U = (L - \omega)/s + 1 \\ M' = M \\ \omega \leq L. \end{cases} \tag{2.1}$$

Given an EEG signal trial completed by the above processing, the adjacent $N = 9$ segments were regarded as one experimental sample. For each EEG segment, we extract the differential entropy (DE) features [33] of five EEG bands ($\delta$ band (1–4 Hz), $\theta$ band (4–8 Hz), low $\alpha$ band (8–11 Hz), high $\alpha$ band (11–14 Hz), $\beta$ band (14–16 Hz)) for each EEG electrode signal.

## 2.4. Personality-based subjects clustering in the macro level

Our goal is to realize cross-subject emotion recognition from EEG signals considering personality. We use k-means clustering algorithm to construct the relationship between subjects according to the following rules:

$$c^{(i)} = \arg \min_{j} \|p^{(i)} - \mu_j\|^2, \tag{2.2}$$

$$\mu_j = \frac{\sum_{i=1}^{P} \delta(c^{(i)}, j) p^{(i)}}{\sum_{i=1}^{P} \delta(c^{(i)}, j)} \tag{2.3}$$

and

$$\delta(c^{(i)}, j) = \begin{cases} 1 & \text{if } c^{(i)} = j \\ 0 & \text{otherwise,} \end{cases} \tag{2.4}$$

where $p^{(i)}$ is the personality of the $i$-th subject, $\mu_j$ ($j = 1, 2, \cdots, P$) is the randomly selected $P$ cluster centroids, and $c^{(i)}$ is the category of the $i$-th subject. Here, $\mu_j$ is updated by equation (2.4).

## 2.5. DNNECA with personality first in the micro level

In this section, we report in detail the EEG channel attention model and the method applied to the cross-subject EEG emotion recognition. Figure 2 illustrates the framework of EEG channel attention model. The raw EEG is preprocessed to extract the DE feature of frequency band. Attention mechanism is aimed at different channels, which have different effects on emotion. Finally, the extracted spatio-temporal information is used as classification feature.

Here, we provide the implementation of the model in detail. As EEG signal is preprocessed, then the data structure is reconstructed, and the frequency band feature is extracted. Let $\mathbf{X} = [x_1, x_2, \ldots, x_N] \in \mathbb{R}^{F \times M \times N}$ be an experimental sample, where $x_i$ is the band feature extracted from the $i$-th segment of the experimental sample, $F$, $M$, $N$ are the number of extracted band features, the number of EEG channels, and the number of EEG segments in an experimental sample, respectively. Initially, we model a neural network layer for each channel to obtain the channel deep information. After learning the channel deep information, the channel-attention layer is used to learn the weight information of channel importance, and then the learned weighted channel information is input into a bidirectional long short-term memory (BiLSTM) network to get the spatial information [18,34]. Finally,

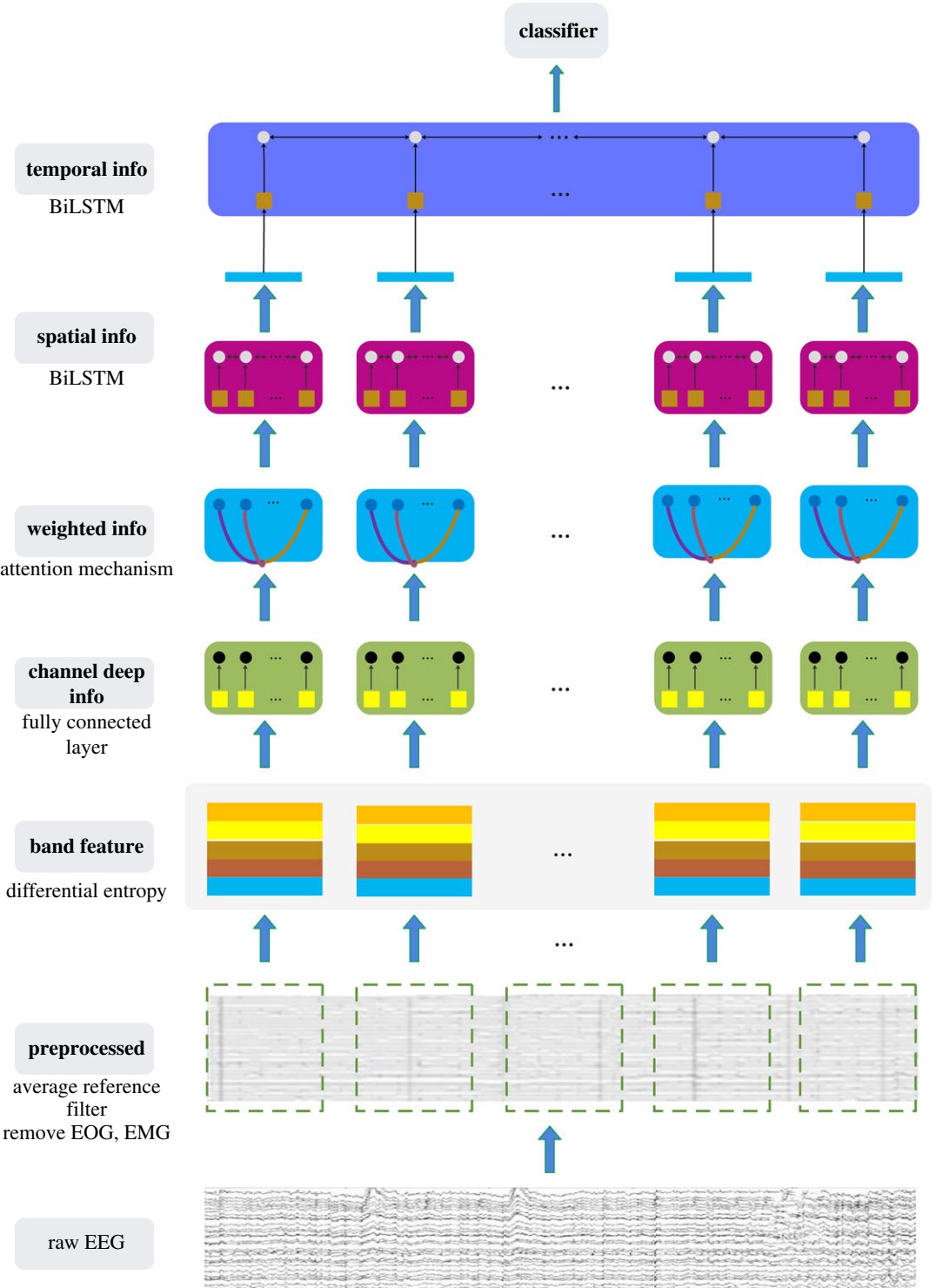

**Figure 2.** Frame diagram based on EEG channel attention model. First, perform corresponding preprocessing on the original data, including clustering operations. Then, EEG spatial information is explored by constructing the relationship between channels, exploring the effects of channels themselves on emotions, and capturing the dynamic information of EEG signals to obtain EEG time information.

the spatial information obtained from each segment is used as input to learn temporal information in BiLSTM, and the final emotion recognition is completed by the classifier. The specific process of these studies is described as follows:

(1) *Channel deep information learning:* For each EEG segment, let $x_i = [\mathbf{m}_{i1}, \mathbf{m}_{i2}, \ldots, \mathbf{m}_{iM}]$ and $\mathbf{m}_{ij}$ denote the frequency band features of $F$ dimension extracted from $j$-th channel, and for segment $x_i$, each column

corresponds to an EEG channel. We model a neural network layer for each channel to obtain the channel deep information. The channel deep information learning can be expressed as

$$\mathbf{a}_{ij} = \mathbf{M}_j \mathbf{m}_{ij} + \mathbf{b}^j, j = 1, 2, \ldots, M \tag{2.5}$$

and

$$\mathbf{A}_i = [\mathbf{a}_{i1}, \mathbf{a}_{i2}, \ldots, \mathbf{a}_{iM}], \tag{2.6}$$

where $\mathbf{M}_j$ indicates learnable transformation matrices and $\mathbf{b}^j$ is the bias.

*(2) Channel weight learning:* Studies have shown that different emotions are closely related to the distribution and selection of EEG channels [19,20]. As the contribution of different EEG channels to emotion recognition is different, we introduce a channel weighting layer to highlight the role of EEG channels in EEG emotion recognition. A channel weight matrix $\mathbf{W} = \{w_{jk}\}$ is chosen to weight each EEG channel, i.e.

$$\mathbf{W} = (\mathbf{Q} \tanh(\mathbf{HA}_i + \mathbf{B}))^T, \tag{2.7}$$

$$w_{jk} = \frac{\exp(w_{jk})}{\sum_{i=1}^{M} \exp(w_{ik})} \tag{2.8}$$

and

$$\hat{\mathbf{A}}_i = \mathbf{A}_i \mathbf{W}, \tag{2.9}$$

where $\mathbf{Q}$ and $\mathbf{H}$ are learnable transformation matrices and $\mathbf{B}$ is the bias matrix. The column of $\mathbf{W}$ is normalized to the weight value by equation (2.8). Obviously, the larger the $w_{jk}$, the more important the $k$-th EEG channel.

*(3) Spatial information learning:* The weighted channel information obtained above is used to capture potential structural information. Here, we use BiLSTM network to capture the spatial information of EEG, which can be formulated as follows:

$$\mathbf{C}_i = \mathcal{BL}(\hat{\mathbf{A}}_i) = [\mathbf{c}_{i1}, \mathbf{c}_{i2}, \ldots, \mathbf{c}_{iM}], \tag{2.10}$$

where $\mathcal{BL}$ is the BiLSTM operation. For the spatial information sequence of EEG, we compress it into a sequence of length $K$ according to the following rules:

$$\hat{\mathbf{c}}_{ik} = \mathrm{ReLU}\left(\sum_{j=1}^{M} g_{jk} c_{ij} + \mathbf{b}_c\right) \tag{2.11}$$

and

$$\hat{\mathbf{C}}_i = [\hat{\mathbf{c}}_{i1}, \hat{\mathbf{c}}_{i2}, \ldots, \hat{\mathbf{c}}_{iK}] \tag{2.12}$$

where $G_c = [g_{jk}]$ is the parameter matrix, $\mathbf{b}_c$ is the bias and ReLU is the activation function.

*(4) Temporal information learning:* We connect the column vectors of $\hat{\mathbf{C}}_i$ into a vector, represented by $\mathbf{d}_i$. The final temporal information matrix $\mathbf{E}$ is calculated as follows:

$$\mathbf{d}_i = [(\hat{\mathbf{c}}_{i1})^{\mathrm{T}}, (\hat{\mathbf{c}}_{i2})^{\mathrm{T}}, \ldots, (\hat{\mathbf{c}}_{iK})^{\mathrm{T}}]^{\mathrm{T}}, \tag{2.13}$$

$$\mathbf{D} = [\mathbf{d}_1, \mathbf{d}_2, \ldots, \mathbf{d}_N] \tag{2.14}$$

and

$$\mathbf{E} = \mathcal{BL}(\mathbf{D}) = [\mathbf{e}_1, \mathbf{e}_2, \ldots, \mathbf{e}_N]. \tag{2.15}$$

*(5) Classifier:* Based on the final temporal information, we use a simple linear transformation to predict the class label of the input EEG experimental sample $\mathbf{X}$, which can be expressed as

$$\mathbf{y} = \mathbf{G}_y \mathbf{e}_N + \mathbf{b}_y = [y_1, y_2, \ldots, y_T], \tag{2.16}$$

where $\mathbf{G}_y$ and $\mathbf{b}_y$, respectively, represent the transformation matrix and bias, and $T$ is the number of classes. The elements in output $\mathbf{y}$ are sent into softmax function for emotional recognition, i.e.

$$P(t \mid \mathbf{X}) = \frac{\exp(y_t)}{\sum_{i=1}^{T} \exp(y_i)}, t = 1, \ldots, T, \tag{2.17}$$

where $P(t \mid \mathbf{X})$ is the probability that input $\mathbf{X}$ is predicted as class $t$.

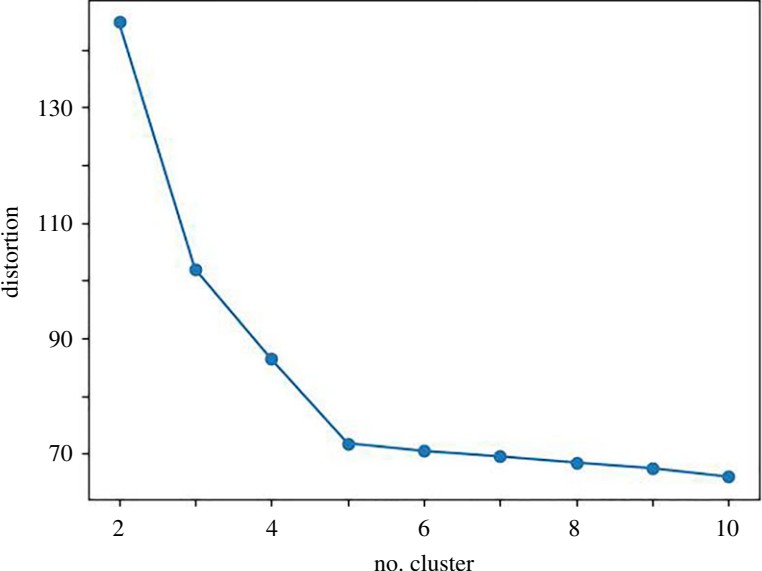

**Figure 3.** Clustering effect of different cluster number. The horizontal axis represents the number of selected clusters, and the vertical axis represents the sum of squares within the cluster.

Suppose that there are training samples, represented by $m$ matrices $\mathbf{X}_i (i = 1, \ldots, m)$. The loss function of the classifier can be expressed as

$$L(\mathbf{X}_i) = -\sum_{t=1}^{m} \tau(l_i, t) \times \log P(t \mid \mathbf{X}_i) \tag{2.18}$$

and

$$J(\mathbf{X}_1, \ldots, \mathbf{X}_m) = \frac{1}{m} \sum_{i=1}^{m} L(\mathbf{X}_i), \tag{2.19}$$

where $l_i$ represents the ground truth label of $\mathbf{X}_i$, which $\tau(l_i, t)$ can be expressed as

$$\tau(l_i, t) = \begin{cases} 1 & \text{if } l_i = t \\ 0 & \text{otherwise}. \end{cases} \tag{2.20}$$

By minimizing the loss function equation (2.19), we can maximize the probability of correctly predicting the emotion class of each training sample. For the test set, we take the category of the maximum probability value as the prediction category of the sample. It is worth noting that the EEG experimental samples trained and tested may come from different subjects when dealing with EEG emotion recognition. In this condition, the emotion recognition model based on training data may not be suitable for test data. In order to realize the cross-subject emotion recognition, we introduce the personality first to divide the subjects into $P$ categories. Moreover, to reduce the individual differences in emotion recognition, an emotion recognition model is trained for each group to make predictions.

## 3. Results and analysis

In order to study the clustering effect of personality, we studied the clustering effect of k-means under different clustering numbers. Figure 3 illustrates an anti-correlation relationship between the number of selected clusters and the sum of squares within the cluster, that is, as the number of cluster increases, distortion decreases. As can be seen, when the number of selected clusters is less than five, the curve descending speed is very fast, but when the number of selected clusters is greater than five, the curve descending speed becomes slower. This indicates that five is the most appropriate cluster number.

Figure 4 presents the relationship between subject number and class when the number of clusters is five. It can be observed from figure 4 that there is basically no clear correlation between subject number and class. Within the cluster are subjects with similar personality, so they may have similar emotions under the same emotional stimulation.

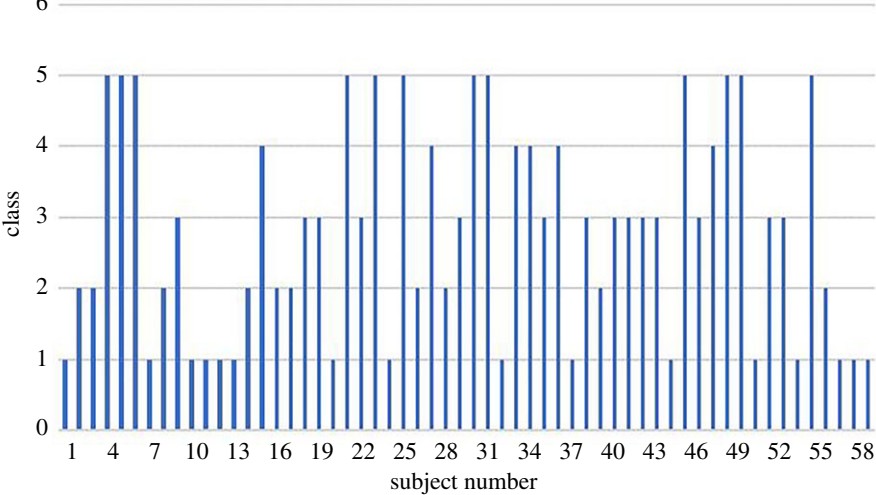

**Figure 4.** Clustering result of the subjects' personality. The horizontal axis represents the number of subject, and the vertical axis represents the category of subjects.

To highlight that the contribution of different EEG channels to emotion recognition is different, we introduce the channel attention layer into the model. The attention matrix of the five groups is shown in figure 5. In addition, the colour of attention matrix represents the contribution of channel. It can be observed that there are differences in the effect of EEG channel on emotion recognition for different personalized population. Similar to [5], we divide it into two parts based on the median value of arousal and valence, and finally use the recognition accuracy as the evaluation standard. We carry out experiments using the channel attention cross-subjects emotion recognition method based on personality first proposed in this paper.

In order to facilitate comparison, we also conduct experiments on emotion recognition without personality first. At the same time, support vector machine (SVM) and XGBoost [35] are used to do the same experiment as our method. During the experiment, we divide the population into different groups, then train and predict each group individually, and finally find the average of the experimental results of all groups. Tenfold cross validation is used for emotion recognition of non-cross subject. For cross-subject emotion recognition, leave-one-subject-out verification strategy is used to evaluate the recognition performance. The test results of non-cross-subject recognition accuracy are shown in table 2. The test results of cross-subject recognition accuracy are shown in table 3.

In general, the results of our proposed method are better than traditional SVM and XGBoost classifiers with or without personality. Meanwhile, in terms of valence and arousal, the accuracy of the three methods based on personality first is higher than emotion recognition without it. Table 2 demonstrates the experimental results in non-cross subjects. As can be seen, without personality first, our proposed method improves performance of SVM and XGBoost by 11.3%, 8.1% on valence, by 9.7%, 7.4% on arousal, and the performance gain of our method (Our) over the method without channel weighting layer (Our-C) is 4.3% on valence, and 4.6% on arousal, respectively. In addition, with personality first, our proposed method improves performance of SVM and XGBoost by 13.7%, 10.4% on valence, by 11.8%, 9% on arousal, and the performance gain of our method (Our) over the method without channel weighting layer (Our-C) is 5.4% on valence, and 5.2% on arousal, respectively.

Table 3 summarizes the experimental results considering cross subjects. As can be seen, without personality first, our proposed method improves performance of SVM and XGBoost by 8.4%, 7% on valence, by 10.6%, 8% on arousal, and the performance gain of our method (Our) over the method without channel weighting layer (Our-C) is 3.3% on valence, and 4.1% on arousal, respectively. Additionally, with personality first, our proposed method improves performance of SVM and XGBoost by 14.8%, 12.5% on valence, by 14.7%, 11.2% on arousal, and the performance gain of our method (Our) over the method without channel weighting layer (Our-C) is 6.2% on valence, and 5.7% on arousal, respectively.

Comparing tables 2 and 3, it is obvious that a significant improvement was obtained in the majority of cases. It is evident that the cross subject emotion recognition based on personality first effectively

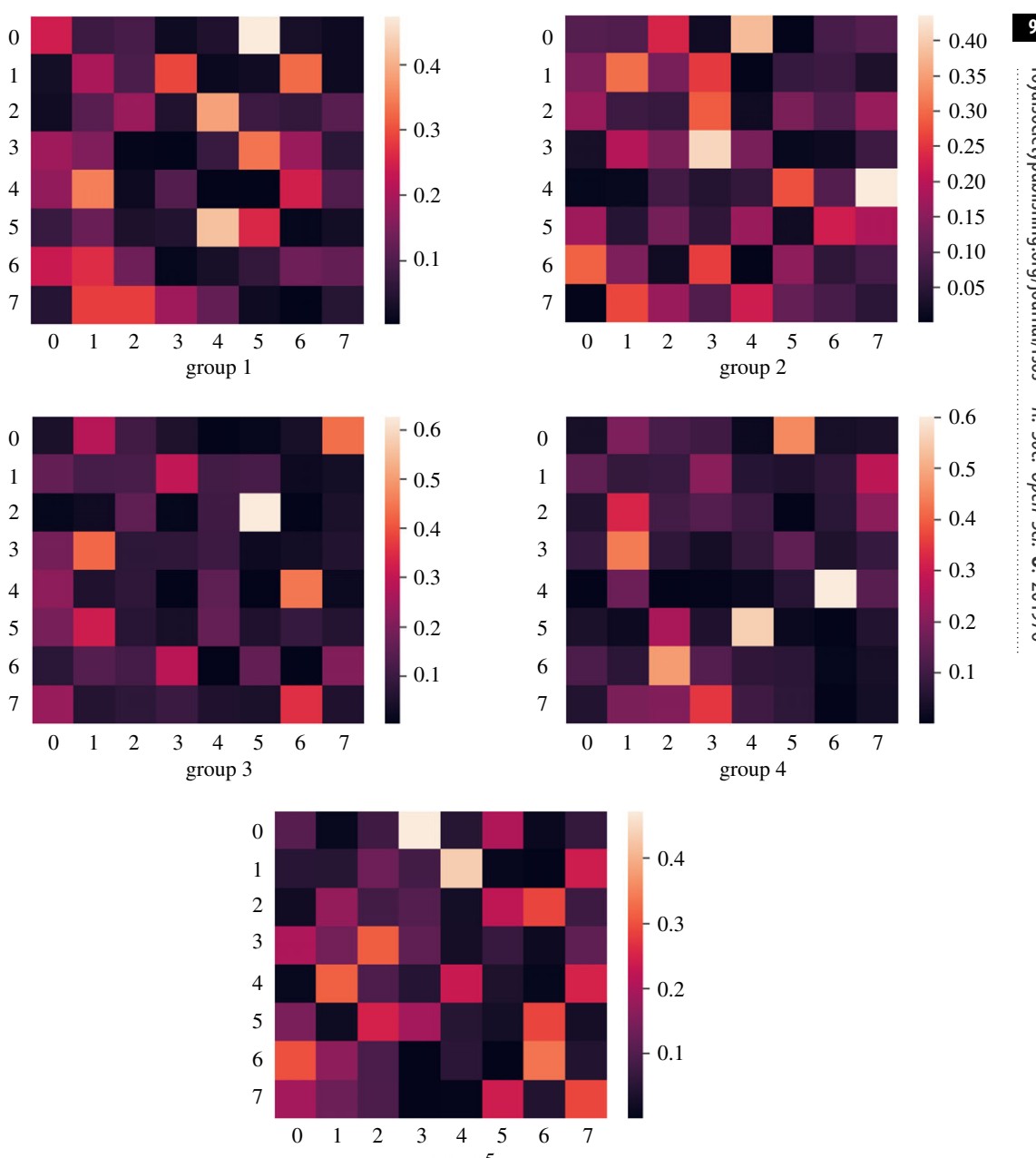

**Figure 5.** Attention matrix of the five groups. Each column corresponds to a channel, and each row corresponds to a time step in BiLSTM. The depth of colour represents the weight of attention. The larger the number of the EEG channel, the greater the contribution rate of the channel to emotion recognition.

**Table 2.** Non-cross-subject emotion recognition results with and without personality first in terms of recognition accuracy (%), where No PF indicates without personality first, where -C indicates without channel weighting layer.

| | no PF | | | | PF | | | |
|---|---|---|---|---|---|---|---|---|
| | SVM | XGB | Our-C | Our | SVM | XGB | Our-C | Our |
| valence | 57.3 | 60.5 | 64.3 | 68.6 | 60.1 | 63.4 | 68.4 | 73.8 |
| arousal | 63.6 | 65.9 | 68.7 | 73.3 | 65.7 | 68.5 | 72.3 | 77.5 |

improves the accuracy of the model, which further proves the effectiveness of the personality first method. We consider personality factors which associate different subjects with similar personality traits. It makes the potential correlation among different subjects effectively mined.

**Table 3.** Cross-subject emotion recognition results with and without personality first in terms of recognition accuracy (%), where No PF indicates without personality first, where -C indicates without channel weighting layer.

| | no PF | | | | PF | | | |
|---|---|---|---|---|---|---|---|---|
| | SVM | XGB | Our-C | Our | SVM | XGB | Our-C | Our |
| valence | 53.8 | 55.2 | 58.9 | 62.2 | 57.6 | 59.9 | 66.2 | 72.4 |
| arousal | 56.2 | 58.8 | 62.7 | 66.8 | 61.2 | 64.7 | 70.2 | 75.9 |

## 4. Discussion and conclusion

With the rise of emotion recognition research, more and more scholars pay attention to the research of EEG channel distribution and channel selection algorithm which are closely related to emotion. Much research has focused on the relationship between personality and emotion [26–28] and studied the reactions to the same emotional stimulus [29]. In this paper, we have considered the personality as a indicator to measure the difference of human emotional response under the same emotional stimulus. Hence, we propose to realize the EEG cross-subject emotion recognition through personality first, and provide a channel attention cross-subject emotion recognition model DNNECA with personality first mechanism. The spatial-temporal EEG information is extracted by considering the weight of channel combined with personality first to complete the cross-subject emotion recognition. Due to the differences between individuals, to effectively carry out the cross-subject emotion recognition, we select individuals with similar personality through personality first, and learn the channel deep information, channel weight information, spatial information and temporal information of them. Additionally, extensive experiments provide compelling evidence that our method is significantly better than that without personality first and can be extended to new subjects with known personality and EEG signals. An important question for future studies is to add some other physiological signals that can evoke emotion, combined with video stimulation content to do personalized emotion recognition. Our methods can be applied to a wide range of areas from emotions, human behaviours to multiplex networks analysis, etc. [10].

Data accessibility. Our article is based on the ASCERTAIN Dataset. If you want to use this data, please refer to the official instructions: http://mhug.disi.unitn.it/wp-content/ASCERTAIN/ascertain.html.

Authors' contributions. Z.T., D.H. and S.Z. did the analytical and numerical calculations. Z.T., D.H. and S.Z. analysed the empirical data. Z.T., Z.Z. and D.J. wrote the manuscript.

Competing interests. The authors declare no conflict of interest. The funders had no role in study design, data collection and analysis, decision to publish or preparation of the manuscript.

Funding. This work was supported by National Natural Science Foundation of China (61902232, 61902231), Natural Science Foundation of Guangdong Province (2019A1515010943), Key Project of Basic and Applied Basic Research of Colleges and Universities in Guangdong Province (Natural Science) (2018KZDXM035), The Basic and Applied Basic Research of Colleges and Universities in Guangdong Province (Special Projects in Artificial Intelligence: 2019KZDZX1030 and 2021A1515012294), the Scientific Research Foundation of Shantou University (grant no. NTF19015) and 2020 Li Ka Shing Foundation Cross-Disciplinary Research (grant nos. 2020LKSFG04D, 2020LKSFG09D).

Acknowledgements. The authors thank anonymous reviewers for their very detailed and helpful review and thanks very much for using ASCERTAIN Dataset.

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
