## [Peer Review File · Royal Society Open Science]

Review History

RSOS-201976.R0 (Original submission)

Review form: Reviewer 1

Is the manuscript scientifically sound in its present form?

Yes

Are the interpretations and conclusions justified by the results?

Yes

Is the language acceptable?

Yes

Do you have any ethical concerns with this paper?

No

Have you any concerns about statistical analyses in this paper?

No

Recommendation?

Accept with minor revision (please list in comments)

Comments to the Author(s)

In view of the specificity of emotion, different subjects may have different subjective responses to the same physical stimulus, so emotion recognition methods based on EEG signals should tend to be personalized. Therefore, the submission starts from the personalization, and proposes a novel personalized EEG emotion recognition from the macro and micro levels. From the perspective of personalization, this paper provides a new idea for EEG emotion recognition.

The method is evaluated on the ASCERTAIN dataset, and the results both in non cross-subject and cross-subject EEG emotion recognition experiments demonstrate that the proposed method can effectively solve the problem of EEG based emotion recognition.

The paper is well written and easy to understand, but I still have some suggestion/comments as follows.

1. This paper presents to use k-means clustering algorithm to construct the relationship between subjects, why we need clustering, why we use k-means method, and what is the motivation of clustering. I think the author should introduce the motivation of clustering in detail.
2. Figure 1 indicates the representation structure of EEG data reconstructed into data unit, I don't think the picture is clear enough. Please give some details about data unit structure.
3. Figure 5 gives the attention matrix of the five groups, from the naked eye, there are some differences, but the author should give the relationship between the attention matrix and personality division, and what are the significant differences between these attention matrixes with forceful, or in other words, give some statistical analysis data or more detailed explanation.

Review form: Reviewer 2

Is the manuscript scientifically sound in its present form?

Yes

Are the interpretations and conclusions justified by the results?

Yes

Is the language acceptable?

Yes

Do you have any ethical concerns with this paper?

No

Have you any concerns about statistical analyses in this paper?

Yes

Recommendation?

Accept with minor revision (please list in comments)

Comments to the Author(s)

The content of this study is that in emotion classification applying machine learning by EEG, the classification accuracy was improved by considering personality compared to not considering it.

EEG-based emotion estimation is currently a field of great interest, and I think the results are also interesting. However, there are still some unclear points, so I hope that you will correct them accordingly.

There were some points that were difficult to understand from model generation to model evaluation, please consider to correct them.

It is difficult to see where the result of clustering is reflected in Figure 2.

In this picture it seems that the clustering was executed in Preprocessed. It is right? it is necessary to mention the clustering in picture and caption.

For 2- (d), it is necessary to describe which data set is used for the clustering. Currently only the explanation of clustering.

For Figure 4

Regarding Figure 4, the authors describe

Within the cluster are subjects with similar personality, so they may have similar emotions under the same emotional stimulation.

However, it was difficult to understand that the result of the group was similar emotions. Originally, the value of the EEG frequency band is not known at all, so it is better to explain the grounds for writing similar emotion in a way that is easy to understand together with the following figure.

Moreover, the author should explain what is the classification result in 2- (d). I think it would be better to show the distributed state of the value of clustering in the figure.

p9

The channel attention layer is difficult to understand. In particular, Figure 5 shows a diagram for each group, but the explanation of the X-axis and Y-axis is not sufficiently explained. It is good to clarify the relationship between position and color in the matrix and attention. In addition, it is necessary to explain the characteristics of each group together with the general interpretation of the EEG frequency band.

p10

Although SVM is mentioned, there is almost no explanation about XGBoost.

It is difficult to understand the meaning of the following sentence, so please explain it. The average recognition accuracy of P groups of people is taken as the final recognition result.

For Table 2, we should state our thoughts on why the Arousal results are better than Valence. Is it because it is the value of EEG?

In table 3,

Is "Our-C" an abbreviation for channel weighted layer? An explanation is needed. It is necessary to describe why the value is lower than that of Our.

The learning result of EEG is an interesting result. However, although EEG is used, the interpretation of frequency bands generally obtained by EEG has not been discussed at all, and it is difficult to understand why it was effective to consider personality clustering. For example, it is expected that analysis of which frequency band of which channel was particularly related to personality will be added in the future.

Decision letter (RSOS-201976.R0)

Dear Dr ZHAO

On behalf of the Editors, we are pleased to inform you that your Manuscript RSOS-201976 "Personality First in Emotion: A Deep Neural Network based on EEG Channel Attention for Cross-Subject Emotion Recognition" has been accepted for publication in Royal Society Open Science subject to minor revision in accordance with the referees' reports. Please find the referees' comments along with any feedback from the Editors below my signature.

Please submit your revised manuscript and required files (see below) no later than 7 days from today's (ie 22-Jun-2021) date. Note: the ScholarOne system will 'lock' if submission of the revision is attempted 7 or more days after the deadline. If you do not think you will be able to meet this deadline please contact the editorial office immediately.

on behalf of Professor Mirella Lapata (Associate Editor) and Marta Kwiatkowska (Subject Editor)
openscience@royalsociety.org

Associate Editor Comments to Author (Professor Mirella Lapata):

Associate Editor: 1

Comments to the Author:

Dear Authors,

I have now obtained the reviews for your manuscript. Both reviewers agree that your paper is worthy of publication, but point out some minor issues which should be fixed. I am therefore accepting your paper with minor revisions. When preparing your final version please make sure to address a) the issues pertaining to your clustering method (motivation); b) address issues

raised by the reviewers pertaining to Figure 2, Figure 1, Figure 5, and Figure 4. c) clarify questions about datasets and methodology as well as explain the attention matrix.

Thank you for submitting your work to our journal!

Best wishes,
Mirella

Reviewer comments to Author:

Reviewer: 1

Comments to the Author(s)

In view of the specificity of emotion, different subjects may have different subjective responses to the same physical stimulus, so emotion recognition methods based on EEG signals should tend to be personalized. Therefore, the submission starts from the personalization, and proposes a novel personalized EEG emotion recognition from the macro and micro levels. From the perspective of personalization, this paper provides a new idea for EEG emotion recognition.

The method is evaluated on the ASCERTAIN dataset, and the results both in non cross-subject and cross-subject EEG emotion recognition experiments demonstrate that the proposed method can effectively solve the problem of EEG based emotion recognition.

The paper is well written and easy to understand, but I still have some suggestion/comments as follows.

1. This paper presents to use k-means clustering algorithm to construct the relationship between subjects, why we need clustering, why we use k-means method, and what is the motivation of clustering. I think the author should introduce the motivation of clustering in detail.
2. Figure 1 indicates the representation structure of EEG data reconstructed into data unit, I don't think the picture is clear enough. Please give some details about data unit structure.
3. Figure 5 gives the attention matrix of the five groups, from the naked eye, there are some differences, but the author should give the relationship between the attention matrix and personality division, and what are the significant differences between these attention matrixes with forceful, or in other words, give some statistical analysis data or more detailed explanation.

Reviewer: 2

Comments to the Author(s)

The content of this study is that in emotion classification applying machine learning by EEG, the classification accuracy was improved by considering personality compared to not considering it. EEG-based emotion estimation is currently a field of great interest, and I think the results are also interesting. However, there are still some unclear points, so I hope that you will correct them accordingly.

There were some points that were difficult to understand from model generation to model evaluation, please consider to correct them.

It is difficult to see where the result of clustering is reflected in Figure 2.

In this picture it seems that the clustering was executed in Preprocessed. It is right? it is necessary to mention the clustering in picture and caption.

For 2- (d), it is necessary to describe which data set is used for the clustering. Currently only the explanation of clustering.

For Figure 4

Regarding Figure 4, the authors describe

Within the cluster are subjects with similar personality, so they may have similar emotions under the same emotional stimulation.

However, it was difficult to understand that the result of the group was similar emotions. Originally, the value of the EEG frequency band is not known at all, so it is better to explain the grounds for writing similar emotion in a way that is easy to understand together with the following figure.

Moreover, the author should explain what is the classification result in 2- (d). I think it would be better to show the distributed state of the value of clustering in the figure.

p9

The channel attention layer is difficult to understand. In particular, Figure 5 shows a diagram for each group, but the explanation of the X-axis and Y-axis is not sufficiently explained. It is good to clarify the relationship between position and color in the matrix and attention. In addition, it is necessary to explain the characteristics of each group together with the general interpretation of the EEG frequency band.

p10

Although SVM is mentioned, there is almost no explanation about XGBoost.

It is difficult to understand the meaning of the following sentence, so please explain it. The average recognition accuracy of P groups of people is taken as the final recognition result.

For Table 2, we should state our thoughts on why the Arousal results are better than Valence. Is it because it is the value of EEG?

In table 3,

Is "Our-C" an abbreviation for channel weighted layer? An explanation is needed. It is necessary to describe why the value is lower than that of Our.

The learning result of EEG is an interesting result. However, although EEG is used, the interpretation of frequency bands generally obtained by EEG has not been discussed at all, and it is difficult to understand why it was effective to consider personality clustering. For example, it is expected that analysis of which frequency band of which channel was particularly related to personality will be added in the future.

===PREPARING YOUR MANUSCRIPT===

Please ensure that you include an acknowledgements' section before your reference list/bibliography. This should acknowledge anyone who assisted with your work, but does not

qualify as an author per the guidelines at <https://royalsociety.org/journals/ethics-policies/openness/>.

===PREPARING YOUR REVISION IN SCHOLARONE===

-- Ensure that your data access statement meets the requirements at <https://royalsociety.org/journals/authors/author-guidelines/#data>. You should ensure that you cite the dataset in your reference list. If you have deposited data etc in the Dryad repository, please only include the 'For publication' link at this stage. You should remove the 'For review' link.

Author's Response to Decision Letter for (RSOS-201976.R0)

See Appendix A.

Decision letter (RSOS-201976.R1)

Dear Dr ZHAO,

I am pleased to inform you that your manuscript entitled "Personality First in Emotion: A Deep Neural Network based on EEG Channel Attention for Cross-Subject Emotion Recognition" is now accepted for publication in Royal Society Open Science.

on behalf of Professor Mirella Lapata (Associate Editor) and Marta Kwiatkowska (Subject Editor)
openscience@royalsociety.org

Appendix A

July 7, 2021

Dear Prof. Mirella Lapata and Marta Kwiatkowska:

Thank you very much for the referee report on our manuscript **RSOS-201976** entitled “Personality First in Emotion: A Deep Neural Network based on EEG Channel Attention for Cross-Subject Emotion Recognition” We are very happy that the referee evaluated our work quite positively and recommended publication in Royal Society Open Science. We have revised the paper to fully address all referee comments. Please find enclosed a detailed, point-to-point response to all referee comments with all changes clearly specified. (The changes in the text are marked.)

We wish to take this opportunity to thank the referee for his/her time and insightful comments that have resulted in an improved manuscript. We would also like to thank you for handling our paper. We hope our revised manuscript can be judged to have met the high standards of Royal Society Open Science.

With kind regards,
Zhidan Zhao
Shantou University
(on behalf of all co-authors)
Email: zzhidanzhao@gmail.com

Point-to-point response to referee comments

1 To Referee 1

The referee stated “ In view of the specificity of emotion, different subjects may have different subjective responses to the same physical stimulus, so emotion recognition methods based on EEG signals should tend to be personalized. Therefore, the submission starts from the personalization, and proposes a novel personalized EEG emotion recognition from the macro and micro levels. From the perspective of personalization, this paper provides a new idea for EEG emotion recognition.

The method is evaluated on the ASCERTAIN dataset, and the results both in non cross-subject and cross-subject EEG emotion recognition experiments demonstrate that the proposed method can effectively solve the problem of EEG based emotion recognition.

The paper is well written and easy to understand, but I still have some suggestion/comments as follows. ” and given some very insightful comments.

Reply: We are grateful that the referee provided a number of comments to improve our paper. In our new version, we have follow the recommendations of the reviewer. Also, your comments have been fully addressed.

Comment 1): This paper presents to use k-means clustering algorithm to construct the relationship between subjects, why we need clustering, why we use k-means method, and what is the motivation of clustering. I think the author should introduce the motivation of clustering in detail.

Reply: Thank you very much for your attention to this fundamental and important issue. First of all, one of the purposes of this article is to study whether the emotions produced by individuals with similar personalities are also similar. The description of personality uses the commonly used Big Five personality model [1]. Secondly, in order to obtain groups with similar personalities, a common method is to aggregate the near ones into one category through the distance between the sample variables, that is, the process of clustering. The commonly used clustering algorithm is the k-means algorithm, which is an unsupervised clustering algorithm, the algorithm process is simple and easy to understand. More importantly, the idea of k-means algorithm is to give a sample set D and divide it into k categories, which is very consistent with the motivation of our manuscript. Therefore, in this manuscript, we use the k-means algorithm for clustering.

Comment 2): Figure 1 indicates the representation structure of EEG data reconstructed into data unit, I don't think the picture is clear enough. Please give some details about data unit structure.

Reply: Thanks for noticing this interesting question. It is difficult to collect medical data of physiological signals such as EEG. More importantly, EEG data involves human privacy issues, so the amount of data is relatively small. However, in deep learning, a large amount of data is needed to participate in the training process. Therefore, when processing EEG data, we sliced the original EEG to achieve the purpose of increasing the amount of data through this common operation, as shown in

Figure 2. This is also the most commonly used method when processing EEG data.

Comment 3): The referee stated “Figure 5 gives the attention matrix of the five groups, from the naked eye, there are some differences, but the author should give the relationship between the attention matrix and personality division, and what are the significant differences between these attention matrixes with forceful, or in other words, give some statistical analysis data or more detailed explanation.”

Reply: Thank you very much for noticing this interesting question. In the analysis of the manuscript, we only observed that the attention matrices of the five groups of people are different based on the analysis results, which shows that for different people, their EEG channels have different contributions to emotion recognition. We also tried to find the relationship between the attention matrix and the crowd, but there is no specific analysis result of these relationships. This question is a more attractive research direction in the future.

2 To Referee 2

The referee stated “The content of this study is that in emotion classification applying machine learning by EEG, the classification accuracy was improved by considering personality compared to not considering it. EEG-based emotion estimation is currently a field of great interest, and I think the results are also interesting. However, there are still some unclear points, so I hope that you will correct them accordingly.

There were some points that were difficult to understand from model generation to model evaluation, please consider to correct them...” and give some insightful suggestions.

Reply: We thank the referee for he/she provided a number of comments to improve our paper, which have been fully addressed, as follows.

Comment 1: It is difficult to see where the result of clustering is reflected in Figure 2. In this picture it seems that the clustering was executed in Preprocessed. It is right? it is necessary to mention the clustering in picture and caption.

Reply: Thank you for noticing this problem and providing valuable suggestions. Yes, the clustering process and Figure 2 are two separate processes. In this article, we first cluster the groups of people according to their personalities, and then for each type of people, use the process of Figure 2 to individually recognize emotions. At the same time, the description of the picture was modified as follows:

1. Section 2, p. 6

“...Frame diagram based on EEG channel attention model. EEG spatial information is explored by constructing the relationship between channels, exploring the effects of channels themselves on emotions, and capturing the dynamic information of EEG signals to obtain EEG time information...”

is revised into,

“...Frame diagram based on EEG channel attention model. First, perform corresponding preprocessing on the original data, including clustering operations. Then, EEG spatial information is explored by constructing the relationship between channels, exploring the effects of channels themselves on emotions, and capturing the dynamic information of EEG signals to obtain EEG time information...”

Comment 2: For Figure 4

Regarding Figure 4, the authors describe within the cluster are subjects with similar personality, so they may have similar emotions under the same emotional stimulation.

However, it was difficult to understand that the result of the group was similar emotions. Originally, the value of the EEG frequency band is not known at all, so it is better to explain the grounds for writing similar emotion in a way that is easy to understand together with the following figure.

Reply: Thank you very much for noticing this basic and interesting question.

We are clustering from the perspective of “Birds of a feather flock together”. We guess that subjects with similar personalities may have similar emotions, which will affect the results of emotion recognition. To a certain extent, our experimental results confirmed this conjecture. Of course, future research should focus more on underlying mechanisms of the above issues.

Comment 3: The referee stated “p9 The channel attention layer is difficult to understand. In particular, Figure 5 shows a diagram for each group, but the explanation of the X-axis and Y-axis is not sufficiently explained. It is good to clarify the relationship between position and color in the matrix and attention. In addition, it is necessary to explain the characteristics of each group together with the general interpretation of the EEG frequency band....”

Reply: Thank you very much for your detailed and professional advice. We learned from the literature that the contribution of EEG channels to emotion recognition is different [2, 3, 4, 5], so we introduced a channel weight layer to express the degree of contribution of different EEG channels to emotion recognition. The larger the number of the EEG channel in Figure 5, the greater the contribution rate of the channel to emotion recognition. Existing studies have confirmed the effectiveness of these EEG frequency bands. In this manuscript, we have extracted the differential entropy [6] features of these frequency bands.

According to your suggestion, we have modified the description of Figure 5 in the revised manuscript. The revised content is as follows:

Section 3, p. 9, Fig. 5

“...Attention matrix of the five groups. Each column corresponds to a channel, and each row corresponds to a time step in BiLSTM. The depth of color represents the weight of attention...”

is revised into,

“...Attention matrix of the five groups. Each column corresponds to a channel, and each row corresponds to a time step in BiLSTM. The depth of color represents the weight of attention. The larger the number of the EEG channel, the greater the contribution rate of the channel to emotion recognition...”

Comment 4: The referee suggested that “ p10 Although SVM is mentioned, there is almost no explanation about XGBoost....”

Reply: Thank you very much for pointing out this basic problem. The XGBoost mentioned in the draft is a common algorithm, and our experimental procedure is similar to that of literature [7]. In the revised manuscript, we have revised the description of this part as follows:

Section 3, p. 9

“...At the same time, Support Vector Machine (SVM) and XGBoost are used ...”

is revised into,

“...At the same time, Support Vector Machine (SVM) and XGBoost [7] are used...”

Comment 5: The referee suggested that “ It is difficult to understand the meaning of the following sentence, so please explain it. The average recognition accuracy of P groups of people is taken as the final recognition result...”.

Reply: Thank you for noticing such a professional and basic problem. In our experimental analysis, for each group of people, they are individually trained and predicted. However, in order to make the results without loss of generality, we finally take the average of these sets of identification data. The modification is as follows:

Section 3, p. 9

“...The average recognition accuracy of P groups of people is taken as the final recognition result... is revised into,

“...During the experiment, we divide the population into different groups, then train and predict each group individually, and finally find the average of the experimental results of all groups...”

Comment 6: The referee stated that “ For Table 2, we should state our thoughts on why the Arousal results are better than Valence. Is it because it is the value of EEG?...”

Reply: Thank you very much for noticing the difference between valence and arousal. Here, valence and arousal represent two indicators for evaluating emotion in the data set, respectively. Among them, valence and arousal are both 7-point scales. However, valence ranges from -3 (very negative) to 3 (very positive), and arousal ranges from 0 (very boring) to 6 (very exciting).

Comment 7: The referee indicated that “In table 3, Is ”Our-C” an abbreviation for channel weighted layer? An explanation is needed. It is necessary to describe why the value is lower than that of Our....”

Reply: Thank you very much for noticing these interesting differences. Yes, ”Our-C” is an algorithm when the channel weight layer is removed in our algorithm. We did this to verify the effectiveness of the channel weight layer, and we also did an experiment after removing the channel weight layer. Here we have revised our explanation as follows:

Section 3, p. 10

“...and the performance gain of our method over the method without channel weighting layer is 5.4% on valence, and 5.2% on arousal respectively...”

is revised into,

“...and the performance gain of our method (Our) over the method without channel weighting layer (Our-C) is 5.4% on valence, and 5.2% on arousal respectively...”

Comment 8: The learning result of EEG is an interesting result. However, although EEG is used, the interpretation of frequency bands generally obtained by EEG has not been discussed at all, and it is difficult to understand why it was effective to consider personality clustering. For example, it is expected that analysis of which frequency band of which channel was particularly related to personality will be added in the future.

Reply: Thank you very much for your constructive suggestion. Due to the limitation of the data set, the correlation analysis between frequency band channel, emotion and personality cannot be obtained. Future research should focus on the correlation analysis between channel and emotion and personality, as well as the correlation between frequency band and emotion.

We wish to take this opportunity to thank the referee for his or her extremely insightful comments and suggestions that helped improve our paper greatly. We also hope that, with the extensive new analysis and computation as well as the new materials provided in the revised manuscript, our manuscript can be judged to have met the high standard of Royal Society Open Science.

References

- [1] L. R. Goldberg, An alternative” description of personality”: the big-five factor structure., *J. Pers. Soc. Psychol.* 59 (6) (1990) 1216.
- [2] S. Jerritta, M. Murugappan, R. Nagarajan, K. Wan, Physiological signals based human emotion recognition: a review, in: 2011 IEEE 7th International Colloquium on Signal Processing and its Applications, IEEE, 2011, pp. 410–415.
- [3] A. Vinciarelli, G. Mohammadi, A survey of personality computing, *IEEE Trans. Affect. Comput.* 5 (3) (2014) 273–291.
- [4] G. Stemmler, J. Wacker, Personality, emotion, and individual differences in physiological responses, *Biol Psychol* 84 (3) (2010) 541–551.
- [5] H. Al Osman, T. H. Falk, Multimodal affect recognition: Current approaches and challenges, *Emotion and Attention Recognition Based on Biological Signals and Images* (2017) 59–86.
- [6] L.-C. Shi, Y.-Y. Jiao, B.-L. Lu, Differential entropy feature for eeg-based vigilance estimation, in: 2013 35th Annual International Conference of the IEEE Engineering in Medicine and Biology Society (EMBC), IEEE, 2013, pp. 6627–6630.
- [7] T. Chen, C. Guestrin, Xgboost: A scalable tree boosting system, in: Proceedings of the 22nd ACM SIGKDD International Conference on Knowledge Discovery and Data Mining, KDD ’16, Association for Computing Machinery, 2016, p. 785–794.